# Corneal Densitometry and In Vivo Confocal Microscopy in Patients with Monoclonal Gammopathy—Analysis of 130 Eyes of 65 Subjects

**DOI:** 10.3390/jcm11071848

**Published:** 2022-03-26

**Authors:** Kitti Kormányos, Klaudia Kovács, Orsolya Németh, Gábor Tóth, Gábor László Sándor, Anita Csorba, Cecília Nóra Czakó, László Módis, Achim Langenbucher, Zoltán Zsolt Nagy, Gergely Varga, László Gopcsa, Gábor Mikala, Nóra Szentmáry

**Affiliations:** 1Department of Ophthalmology, Semmelweis University, Mária utca 39, 1085 Budapest, Hungary; kovacs.klaudia94@gmail.com (K.K.); gabortothgabor@gmail.com (G.T.); sandorgaborlaszlo@gmail.com (G.L.S.); csorbani@gmail.com (A.C.); cecilia.czako@gmail.com (C.N.C.); zoltan.nagy100@gmail.com (Z.Z.N.); nora.szentmary@uks.eu (N.S.); 2Department of Ophthalmology, Markusovszky University Teaching Hospital, 9700 Szombathely, Hungary; nemeth.orsolya22@gmail.com; 3Department of Ophthalmology, Faculty of Medicine, University of Debrecen, 4032 Debrecen, Hungary; modisjr@gmail.com; 4Experimental Ophthalmology, Saarland University, 66421 Homburg/Saar, Germany; achim.langenbucher@uks.eu; 53rd Department of Internal Medicine and Haematology, Semmelweis University, 1085 Budapest, Hungary; vargager@gmail.com; 6Department of Haematology and Stem Cell-Transplantation, South-Pest Central Hospital—National Institute for Hematology and Infectious Diseases, 1097 Budapest, Hungary; laszlogopcsa@yahoo.com (L.G.); gmikala67@gmail.com (G.M.); 7Dr. Rolf M. Schwiete Center for Limbal Stem Cell and Congenital Aniridia Research, Saarland University, 66424 Homburg/Saar, Germany

**Keywords:** monoclonal gammopathy, MGOS, cornea, Pentacam, IVCM

## Abstract

Background: Corneal imaging may support an early diagnosis of monoclonal gammopathy. The goal of our study was to analyze corneal stromal properties using Pentacam and in vivo confocal cornea microscopy (IVCM) in subjects with monoclonal gammopathy. Patients and methods: In our cross-sectional study, patients with monoclonal gammopathy (130 eyes of 65 patients (40.0% males; age 67.65 ± 9.74 years)) and randomly selected individuals of the same age group, without hematological disease (100 eyes of 50 control subjects (40.0% males; age 60.67 ± 15.06 years)) were included. Using Pentacam (Pentacam HR; Oculus GmbH, Wetzlar, Germany), corneal stromal light scattering values were obtained (1) centrally 0–2 mm zone; (2) 2–6 mm zone; (3) 6–10 mm zone; (4) 10–12 mm zone. Using IVCM with Heidelberg Retina Tomograph with Rostock Cornea Module (Heidelberg Engineering, Heidelberg, Germany), the density of hyperreflective keratocytes and the number of hyperreflective spikes per image were manually analyzed, in the stroma. Results: In the first, second and third annular zone, light scattering was significantly higher in subjects with monoclonal gammopathy, than in controls (*p* ≤ 0.04). The number of hyperreflective keratocytes and hyperreflective spikes per image was significantly higher in stroma of subjects with monoclonal gammopathy (*p* ≤ 0.012). Conclusions: Our study confirms that increased corneal light scattering in the central 10 mm annular zone and increased keratocyte hyperreflectivity may give rise to suspicion of monoclonal gammopathy. As corneal light scattering is not increased at the limbal 10–12 mm annular zone in monoclonal gammopathy subjects, our spatial analysis provides evidence against the limbal origin of corneal paraprotein deposition. Using IVCM, stromal hyperreflective spikes may represent specific signs of monoclonal gammopathy.

## 1. Introduction

Monoclonal gammopathy comprises a broad spectrum of diseases, ranging from monoclonal gammopathy of undetermined significance (MGUS) to multiple myeloma (MM) [1,2,3].

In the case of monoclonal gammopathy (MG), monoclonal proteins may be deposited in various organs [4,5,6,7,8,9,10,11,12], resulting in monoclonal gammopathy of clinical significance (MGCS) [13]. If paraprotein deposition occurs exclusively in the eye, the term “monoclonal gammopathy of ocular significance” (MGOS) is used [14].

Between the ocular manifestation of monoclonal gammopathy, myositis, proptosis [15], conjunctival and corneal deposits, acute or chronic uveitis [16,17], maculopathy, foveolar drusen [18,19,20], Doyne retinal dystrophy [21] and central retinal artery or vein occlusion [22] have been described.

Corneal deposits associated with monoclonal gammopathy were first described in 1934 by Meesman [23]. These were later described as chameleon-like changes and have been named paraproteinemic keratopathy (PPK) [24,25]. These deposits are white, yellow, brown or grey in color and may be found in any layer of the cornea [26]. In 2012, Lisch et al. created a nomenclature distinguishing five different types of immunotactoid keratopathy (ITK): crystalline-like ITK, lattice-like ITK, peripheral granular-like ITK, peripheral band-like ITK, and peripheral patch-like ITK [27]. In 2016, they expanded their classification to 11 MGUS-induced paraproteinemic keratopathy forms [24].

Corneal properties may be objectively analyzed using a slit lamp, corneal topography or tomography, optical coherence tomography (OCT), ultrasound biomicroscopy and in vivo confocal microscopy.

Scheimpflug-based densitometry of the anterior segment is becoming increasingly important in corneal diagnostics, and it is widely used in everyday clinical practice. The most important properties of the healthy cornea are clarity and transparency. Scheimpflug imaging analysis can be used to measure light transmission and backscatter [28,29]. Studies have shown that even in cases of clinically clear corneas, there may be a greater degree of corneal backscatter [30]. The Oculus Pentacam (Oculus Inc., Oculus GmbH, Wetzar, Germany), as a corneal tomographer, employs the Scheimpflug principle to obtain images of the anterior segment. The rotating camera captures 25 anterior segment images in 2 s, thus providing a quantifiable measurement of corneal clarity. In addition to the use of Pentacam in routine clinical diagnostics as a tomographer, using Pentacam, corneal light scattering has been extensively studied in a number of ocular diseases. There is an increased corneal light scattering in keratoconus [31], vernal keratoconjunctivitis [32], corneal dystrophies [33], cornea guttata [34], infectious corneal infiltrates [35], following penetrating and lamellar keratoplasty [36], following refractive corneal procedures [37] and after collagen crosslinking (CXL) [38]. Nevertheless, corneal light scattering is decreased in highly myopic corneas and in diabetic patients [39,40]. In 2017, in a retrospective study by Enders at al. summarized the capability of Scheimpflug-based densitometry of the cornea, to quantify light chain deposits in five patients with monoclonal gammopathy [41]. In 2017, Busch et al. analysed 20 eyes [42] and later, in 2019, Ichii et al. examined 30 subjects with monoclonal gammopathy, also using Pentacam [43].

In vivo confocal microscopy (IVCM) represents another objective examination method to evaluate corneal morphology and to assess layer reflectivity and cellular density. Corneal deposits, occurring in monoclonal gammopathy, were first described in single case reports using IVCM [25,44,45,46,47,48,49,50,51,52,53]. Thereafter, in a cross-sectional study, Aragona et al. examined 31 patients with MGUS, smouldering myeloma and MM, using IVCM [54]. All these studies described hexagonal or round deposits with a crystalline appearance in the corneal stroma.

Nevertheless, no previous study has analyzed and compared corneal stromal properties using Pentacam and IVCM at the same time, in monoclonal gammopathy. Although Enders et al. analysed 5 [41], Busch et al. 20 [42] Ichii et al. 30 subjects with monoclonal gammopathy using Pentacam [43], and Aragona et al. analyzed 31 patients with IVCM [54], none of these studies analyzed a larger cohort of subjects, in a cross-sectional manner.

The purpose of our present study was to analyze and compare corneal stromal light scattering using Pentacam and corneal properties using IVCM in subjects with monoclonal gammopathy and in controls, in a cross-sectional manner.

## 2. Patients and Methods

In our cross-sectional study, patients of the Department of Hematology and Stem Cell-Transplantation of the South-Pest Center Hospital—National Institute for Hematology and Infectious Disease, Budapest, Hungary and the 3rd Department of Internal Medicine and Haematology, Semmelweis University, Budapest, Hungary, diagnosed and treated with monoclonal gammopathy between 1999–2021 have been included. As a control group, randomly selected individuals of the same age group, without haematological disease have been included. The local Ethics Committee gave permission to our study (OGYÉI/50115/2018). Participation in this study was voluntary, and written informed consent was obtained from all participants.

We analyzed altogether 230 eyes of 115 patients (40% males; age 64.96 ± 12.28 (33–86) years). There were 130 eyes of 65 patients (40.0% males; age 67.71 ± 9.40 (range 38–83) years) with monoclonal gammopathy (MG) and 100 eyes of 50 subjects (40.0% males; age 60.67 ± 15.06 (range 33–86) years), as controls. The ages of the patients with MG and controls did not differ significantly (*p* = 0.267).

In patients with established hematological diagnosis, the time of the hematological diagnosis was in one case (1.54%) within 1 year, in 28 (43.08%) cases within 5 years, in 32 (49.23%) cases within 5–10 years and in 4 (6.15%) cases more than 10 years ago. The hematological diagnosis was MGUS in 6 (9.23%), multiple myeloma in 50 (76.92%), smoldering myeloma, amyloidosis or Waldenström macroglobulinemia in 3-3-3 cases (4.61%-4.61%-4.61%).

With respect to immunoglobulin heavy chains, there was an increased IgG level in 39 individuals (60%), an increased IgA level in 15 (23.08%), an increased IgM level in 6 (9.23%), and an increased IgD level in 1 (1.54%) case. In 1 (1.54%) case we found biclonal elevation of IgG and IgM heavy chains. Considering light chains, in 40 (61.54%) subjects kappa chain, and in 25 (38.46%) patients lambda chain was verified and in 2 cases (3.08%) aberrant heavy chain production was not detectable.

Before ophthalmic examination of MG subjects and controls, ophthalmic medical history was taken. Thereafter, ophthalmic examination included a visual acuity test using trial glasses in a trial frame (best corrected visual acuity), slit-lamp examination following dilation of the pupil, Scheimpflug imaging (Pentacam HR; Oculus GmbH, Wetzlar, Germany) and in vivo confocal laser scanning cornea microscopy using the Heidelberg Retina Tomograph with Rostock Cornea Module (HRTII/RCM) (Heidelberg Engineering, Heidelberg, Germany).

Using Pentacam, keratometric values, corneal astigmatism and corneal apex pachymetry were measured automatically by the software and these data were collected. In addition, corneal backscattered light values in grey scale unit (light scattering) were recorded from 0 (100% transparent) to 100 (completely opaque, 0% transparent) [41]. For analysis of the data, we used the corneal densitometry average table, according to Enders at al. [41]. Values were obtained in 4 annular zones of the cornea, which were centered to the apex of the cornea: (1) central annular 0–2 mm zone; (2) intermediate 2–6 mm zone; (3) peripheral 6–10 mm zone; (4) limbal 10–12 mm zone. Additionally, these annular zones were divided into the following 3 corneal stromal layers according to their depth: (A) anterior 120 µm deep corneal stromal layer (AL), (B) middle corneal stromal layer more than 120 µm from the anterior and less than 60 µm from the posterior corneal stromal surface (ML) and (C) posterior corneal stromal layer (PL), less than 60 µm from the posterior corneal stromal surface. The total corneal stromal volume (between the epithelium and endothelium) was also analyzed (TL).

Before scanning with the in vivo confocal laser scanning cornea microscope, one drop of 0.4% oxybuprocaine hydrochloride (Novesine, OmniVision GmbH, Puchheim, Germany) was instilled in the conjunctival sac, as an anesthetic. As a coupling medium to ensure the airless contact between the plastic cap, covering the immersion lens of the microscope (a sterile poly-methyl-methacrylate cap (TomoCap; Heidelberg Engineering, Heidelberg, Germany) and the ocular surface, one drop of artificial tear gel (0.2% carbomer, Vidisic, Dr Mann Pharma, Berlin, Germany, Bausch&Lomb) was instilled.

Two dimensional images were captured in every corneal layer from the epithelium to the endothelium by the instrument’s section mode. These images represent an “en face” section of the cornea with a resolution of 384 × 384 pixels covering a 400 µm × 400 µm area. The depth of the examination field in the cornea was ensured by the inbuilt digital micrometer gauge. A diode laser beam with a wavelength of 670 nm was used by the HRTII/RCM to scan the focal plane of the examined specimen. The same examiner (KK) recorded and analyzed the captured micrographs. Two well focused images were randomly selected in each corneal layer for detailed analysis. For description of the data, we used the classification of Aragona et al. [54], with some modifications. We extended the analysis of Aragona et al. with description of corneal stromal properties in the anterior (anterior 120 µm deep corneal stromal layer), middle (middle corneal stromal layer more than 120 µm from the anterior and less than 60 µm from the posterior corneal stromal surface) and posterior corneal stromal layers (less than 60 µm from the posterior corneal stromal surface) (Table 1).

In order to analyze hyperreflectivity of the corneal epithelial cells per micrograph, we used the following arbitrary scoring system (Table 1, Figure 1): “1” = no hyperreflective epithelial cells (no alteration), “2” ≤ 4 hyperreflective epithelial cells (mild alteration), “3” > 4 hyperreflective epithelial cells (moderate alteration), “4” = no images for evaluation.

Describing keratocytes, we used the following arbitrary scoring system (Table 1, Figure 1): “0” = no changes, “1” ≤ 4 hyperreflective keratocytes per micrograph, “2” = 5–7 hyperreflective keratocytes per micrograph, “3” = 8–16 hyperreflective keratocytes per micrograph, “4” > 16 hyperreflective keratocytes per micrograph, “5” no images for evaluation in the questioned layer.

Analysis of stromal hyperreflective spikes was carried out according to the following arbitrary system (Table 1 and Figure 1): “0” = no changes, hyporeflective matrix, “1” = hyperreflective areas, maximal 1 spike per micrograph, “2” = 2–3 spikes per micrograph,”3” ≥ 4 spikes per micrograph, “4” = giant spike/s (>75 µm), “5” = no images for evaluation.

Description of the endothelial cell layer included the following information (Table 1, Figure 1): “1” = no changes, “2” = hyperreflective changes; “3” = guttae, “4” = no images for evaluation.

For statistical analyzis of the data, the Mann–Whitney U test and the χ^2^ test were used, with *p* values below 0.05 considered statistically significant.

## 3. Results

BCVA did not differ significantly between subjects with monoclonal gammopathy (0.83 ± 0.25 (0.01–1.0) (logMAR 0.1 ± 0.24)) and controls (0.92 ± 0.21 (0.06–1.0) (logMAR 0.1 ± 0.21)) at the examination time-point (*p* = 0.912). Keratometric values, corneal astigmatism and corneal apex pachymetry also did not show a significant difference between both groups (Table 2) (*p* ≤ 0.724).

Using slitlamp examination, there were corneal opacities (corresponding to paraproteinemic keratopathy) in 12 (9.23%) eyes of 8 (12.31%) subjects with monoclonal gammopathy.

Using Pentacam, in the first, second and third annular zone and along all analyzed corneal zones together, including anterior, middle and posterior corneal stromal layers, light scattering was significantly higher in monoclonal gammopathy subjects, than in controls (*p* ≤ 0.04). Nevertheless, in the fourth annular zone (10–12 mm), corneal light scattering did not differ between groups (*p* ≥ 0.152) (Figure 2).

Using IVCM, epithelial cell layer hyperreflectivity was significantly higher in controls, than in MG subjects (*p* < 0.001) (Table 3, Figure 3). With IVCM, the number of stromal hyperreflective keratocytes per micrograph was significantly higher in anterior, middle and posterior stromal layers of subjects with monoclonal gammopathy, than in controls (*p* < 0.001) (Table 3, Figure 3). In MG subjects, a higher proportion of subjects belonged to groups 3, 4 and 5, as in controls, concerning stromal hyperreflective keratocytes. The number of stromal hyperreflective spikes per micrograph was also significantly higher in anterior, middle and posterior stromal layers of subjects with monoclonal gammopathy, than in controls (*p* ≤ 0.015). Concerning spikes, proportion of MG subjects was higher in groups 3 and 4 than those in controls (Table 3, Figure 3). Using in vivo confocal microscopy, endothelial cell layer properties did not differ significantly between MG subjects and controls (*p* = 0.059, Table 3, Figure 3).

## 4. Discussion

Paraproteinemic keratopathy is a relative rare ocular sign of monoclonal gammopathy. Most ophthalmologists do not recognize paraproteinaemic keratopathy and do not send symptomatic subjects to hematological examination. Garibaldi et al. summarized previous case reports and case series from the literature in 2005 [25]. Using slitlamp biomicroscopy, Bourne et al. [55] described corneal opacities in 1 of 100 monoclonal gammopathy subjects. Arson and Shaw [56] could not verify corneal involvement in 13 subjects with multiple myeloma using a slitlamp. In our present study, we observed paraproteinemic keratopathy in 12 (9.23%) eyes, using a slitlamp and analyzing a larger cohort of 130 eyes of patients with monoclonal gammopathy. This higher percentage of subjects with corneal involvement could be explained through the longer standing (in most of the cases 5–10 years) hematological disease of the patients. Nevertheless, paraproteinemic keratopathy did not influence keratometric values, corneal astigmatism and central corneal thickness in patients of the present study.

Using Pentacam, corneal light scattering was significantly higher in the anterior, central and posterior stromal layers of the cornea in the central, intermediate and peripheral annular corneal zones (all together 0–10 mm centrally) of monoclonal gammopathy patients, than in healthy controls. None of the patients had corneal pathologies or previous corneal surgeries which may have resulted in an increased corneal light scattering. An increased corneal thickness may also result in increased corneal light scattering. Nevertheless, as corneal thickness (apex pachymetry) did not differ between MG subjects and controls (Table 2), the increased corneal light scattering could not be related to this factor.

Busch and Ichii [42,43] also reported a significantly higher corneal light scattering at the central 6 mm diameter corneal area in the anterior and central stroma of patients with monoclonal gammopathy, analyzing 10 and 30 MG subjects. Enders et al. [41] found a significantly increased corneal light scattering in the central 10 mm corneal zone along the total corneal thickness (anterior, central and posterior stroma), in five patients with monoclonal gammopathy. Our study provides additional strength to previous studies showing increased corneal light scattering in 130 eyes with monoclonal gammopathy—in a larger cohort of subjects (Table 4). Therefore, in case other listed corneal pathologies may be excluded, an increased central corneal light scattering (0–10 mm central corneal zone) may arise suspicion of corneal changes due to monoclonal gammopathy.

Nevertheless, there may be several changes in the corneal stroma of MG subjects over time. With increasing disease length, corneal stromal deposition may increase, although, this may also decrease again in the case of systemic treatment of the hematological disease. These processes may all bear an impact on corneal light scattering over time. In our opinion, an increased corneal light scattering may be an important sign of monoclonal gammopathy, nevertheless, describing the course of the disease, it may be more appropriate to describe an impaired light scattering in MG, as it has also been described in the publication of Ichii et al. [43]. Corneal light scattering changes in the course of diseases with monoclonal gammopathy needs further analysis.

The origin of the corneal deposits in monoclonal gammopathy is still not well understood. Some authors suggest that these may be delivered from the limbal vessels to the cornea [57] or may be transported from the tear film (immunoglobulins), or from the aqueous humor [18,25]. Some authors also suggested that these deposits may be locally synthesized through stromal keratocytes [25]. Based on the results of our densitometric analysis, direct immunoglobulin transport from the limbal vessels is less likely, as we did not find an increased corneal light scattering at the corneal limbal zone of MG subjects, compared to controls. We also could not find a predominantly anterior or posterior stromal increase of corneal light scattering in MG corneas, referring to a potential anterior or posterior origin of corneal deposits from the tear film (anteriorly) or from the aqueous humor (posteriorly).

In vivo confocal microscopy is a noninvasive device to imagine the cornea at the cellular and microstructural level. Its application has expanded over the past decades. Nevertheless, using IVCM, several corneal pathologies should be recognized or excluded, in order to avoid misinterpretation of the images.

With IVCM, the number of stromal hyperreflective keratocytes and hyperreflective spikes per micrograph was significantly higher in anterior, middle and posterior stromal layers of subjects with monoclonal gammopathy, than in controls (*p* ≤ 0.012). In contrast, Aragona et al. [54] described a significantly decreased keratocyte density in subjects with MG, examining 31 patients with MG, in Messina, Italy. Keratocyte density may be increased in keratitis [58,59,60,61,62], in autoimmune diseases [63], in some corneal dystrophies [58,64,65,66,67,68], or following corneal surgeries such as crosslinking or corneal transplantation [69,70,71,72,73,74]. Nevertheless, keratocyte density decreases in ectatic corneal diseases [75,76,77,78,79], and congenital glaucoma [80]. None of the analyzed subjects had any of these diagnoses in the present study. Therefore, we speculate that the increased keratocyte density is rather related to the hematological disease of the patients. Corneal stromal cells may behave similarly to hematopoietic stem cells [81]. The phenomenon, that these may undergo myofibroblastic transformation is well known [82]. In addition, these may have a very similar gene expression profile to bone marrow cells [83]. Therefore, it is possible, that the keratocyte activation in these patients may show parallel features to bone marrow changes/activity of the MG subjects. This needs further analysis.

Beside one case report [84], no previous study described the appearance of hyperreflective spikes in the corneal stroma of monoclonal gammopathy subjects. Generally, as an example, amyloid, chloroquine, ciprofloxacin, gold and iron may all cause stromal deposition [58,84,85,86]. On the other hand, these do not result in hyperreflective stromal spikes in confocal microscopy [16,25,87]. Subepithelial nerves may be falsely interpreted as hyperreflective spikes using IVCM. Nevertheless, these are not present in an increased density in the deeper corneal stromal layers, which therefore helps in an appropriate interpretation of IVCM images. To our knowledge, an increased nerve density in the middle and posterior corneal stromal layers of monoclonal gammopathy subjects has not yet been described. We suggest that the detected hyperreflective stromal spikes may either show corneal stromal immunoglobulin deposition (invisible with the slit lamp but detectable using IVCM) or may be present due to stromal drug deposition (systemic treatment of the hematological disease). These also need further clarification.

In summary, our study confirms that increased corneal light scattering in the central 10 mm annular zone and increased keratocyte hyperreflectivity may give rise to suspicion of monoclonal gammopathy. As corneal light scattering is not increased at the limbal 10–12 mm annular zone of monoclonal gammopathy subjects, our spatial analysis provides evidence against the limbal origin of corneal paraprotein deposits. Using IVCM, stromal hyperreflective spikes may represent specific signs of monoclonal gammopathy, independent of the depth of their stromal localization. Nevertheless, during follow-up of a hematological disease, corneal stromal changes must be further analyzed to obtain better insight into their pathophysiology and in corneal symptoms of monoclonal gammopathy.

## Figures and Tables

**Figure 1 jcm-11-01848-f001:**
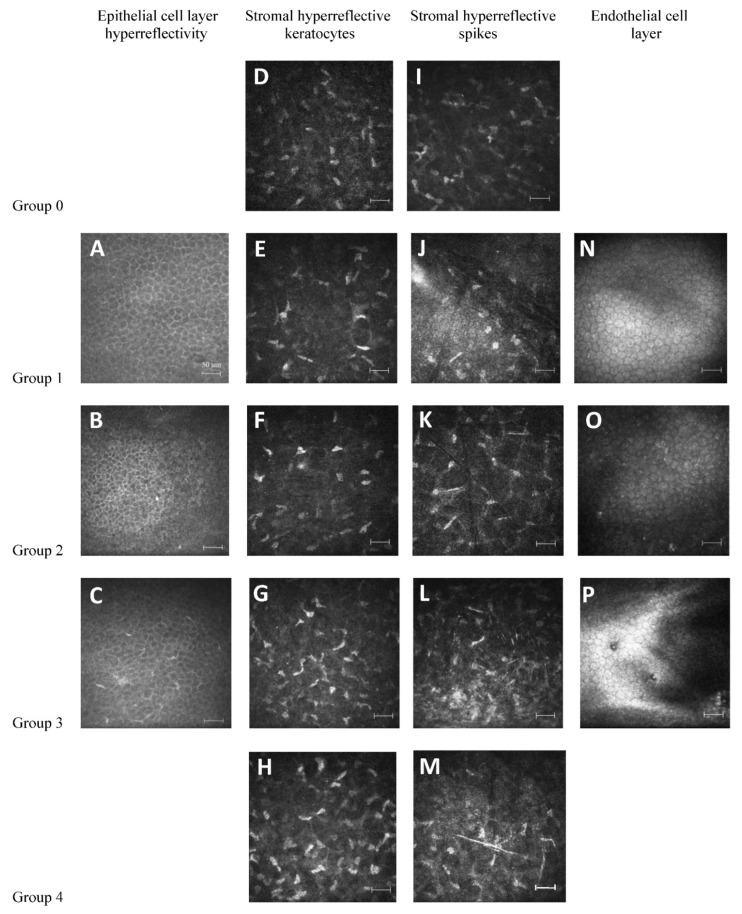
Classification of epithelial cell layer properties (**A**–**C**), stromal hyperreflective keratocytes (**D**–**H**), stromal hyperreflective spikes (**I**–**M**) and endothelial cell layer properties (**N**–**P**) using vivo confocal laser scanning cornea microscopy (IVCM) imaging with Heidelberg Retina Tomograph, with Rostock Cornea Module (HRTII/RCM) (Heidelberg Engineering, Heidelberg, Germany), as described at Table 1. We used a classification modified from Aragona et al. [ [54]. Scale bars: 50 µm.

**Figure 2 jcm-11-01848-f002:**
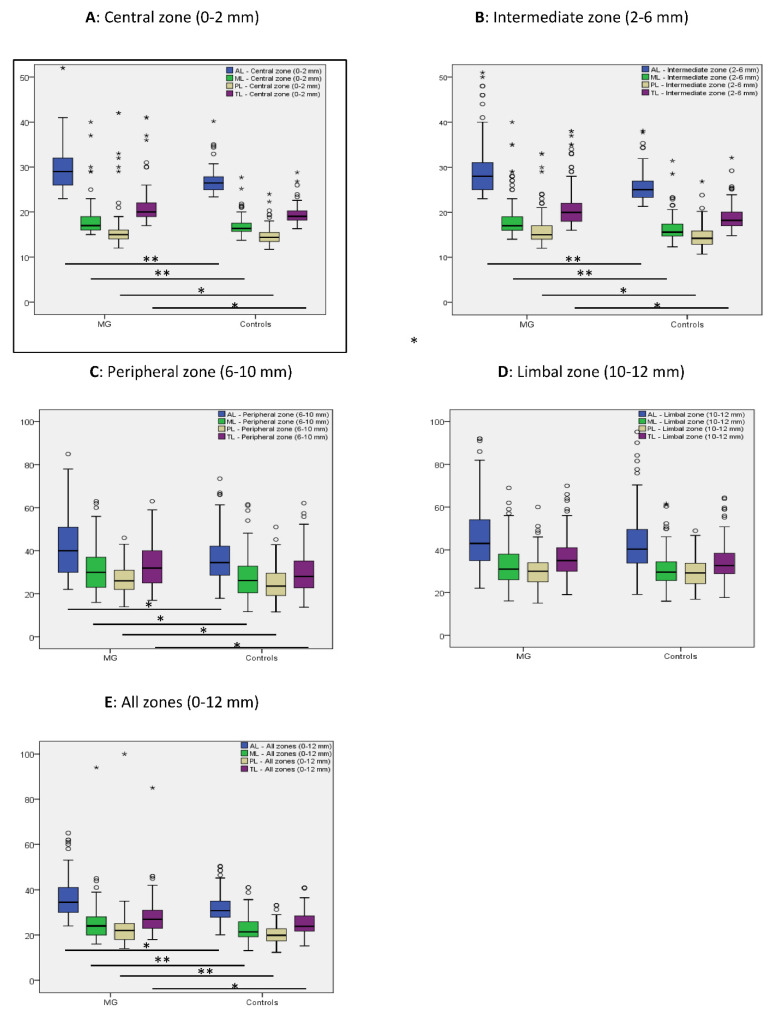
Corneal densitometry values at the central (**A**), intermediate (**B**), peripheral (**C**) and limbal (**D**) annular corneal zones and in all stromal zones (0–12 mm) (**E**) using Pentacam (Oculus, Wetzlar, Germany) at the anterior (AL), middle (ML) and posterior (PL) corneal stromal layers and along the complete corneal thickness (TL) in monoclonal gammopathy (MG) and in control subjects. Small circles in the graphs show extreme values and stars outliers. *p* values show results of the statistical analysis using Mann–Whitney U test. *p* values below 0.001 are marked with “**”, other significant *p* values are marked with “*”.

**Figure 3 jcm-11-01848-f003:**
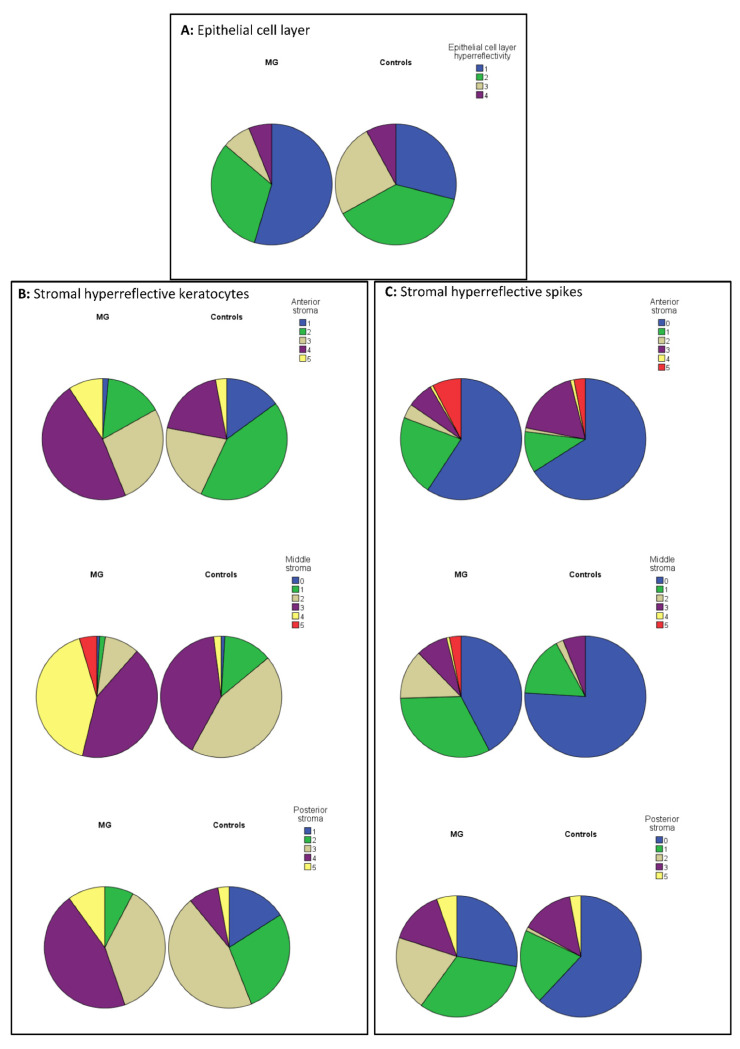
Proportion of eyes with monoclonal gammopathy and proportion of control eyes in groups 0–5, concerning epithelial cell layer properties (**A**), hyperreflective stromal keratocytes per micrograph (**B**), hyperreflective stromal spikes per micrograph (**C**) and endothelial cell layer properties (**D**). The arbitrary scoring system, described in Table 1, has been used.

**Table 1 jcm-11-01848-t001:** Classification of epithelial cell layer properties, stromal hyperreflective keratocytes, stromal hyperreflective spikes and endothelial cell layer properties. We used a classification modified from Aragona et al. [54].

	Epithelial Cell Layer Hyperreflectivity/Micrograph	Number of Stromal Hyperreflective Keratocytes/Micrograph	Number of Stromal Hyperreflective Spikes/Micrograph	Endothelial Cell Layer
0	-	no changes	no changes	-
1	no changes	≤4	≤1	no changes
2	≤4	5–7	2–3	hyperreflective changes
3	>4	8–16	≥4	guttae
4	no images for evaluation	>16	giant spike/s (>75 µm)	no images for evaluation
5	-	no images for evaluation	no images for evaluation	-

**Table 2 jcm-11-01848-t002:** Keratometric values and corneal astigmatism in Diopters (D), axis of astigmatism and apex pachymetry in subjects with monoclonal gammopathy (MG) and in controls, using Pentacam (Oculus, Wetzlar, Germany). *p* values show results of the statistical analysis using Mann–Whitney U test.

	K1 (D)	K2 (D)	Corneal Astigmatism (D)	Axis of Corneal Astigmatism (Degree)	Apex Pachymetry (µm)
MG	43.25 ± 1.63 (35.70–47.50)	44.24 ± 1.71 (40.50–52.50)	0.99 ± 1.06 (0.00–9.70)	0.51 ± 0.23 (−0.66–0.98)	569.71 ± 144.04 (465–2662)
Controls	43.22 ± 1.64 (40.20–47.50)	44.07 ± 1.70 (41.00–48.00)	0.84 ± 0.59 (0.10–3.50)	0.49 ± 0.23 (−0.56–0.98)	560.51 ± 36.21 (465–634)
*p* value	0.255	0.098	0.127	0.137	0.724

**Table 3 jcm-11-01848-t003:** Number (%) of eyes with monoclonal gammopathy and number (%) of control eyes in groups 0–5, concerning epithelial cell layer properties, number of hyperreflective stromal keratocytes and hyperreflective stromal spikes per micrograph and endothelial cell layer properties. *p* values show results of the statistical analysis using χ^2^ test. The arbitrary scoring system, described in Table 1, has been used. * Significantly higher in controls, as in MG subjects.

		Epithelial Cell Layer Hyperreflectivity/Micrograph	Number of Stromal Hyperreflective Keratocytes/Micrograph			Number of Stromal Hyperreflective Spikes/Micrograph			Endothelial Cell Layer
			Anterior Stroma	Middle Stroma	Posterior Stroma	Anterior Stroma	Middle Stroma	Posterior Stroma	
**MG**	0	-	0	1 (0.8%)	0	77 (59.2%)	55 (42.3%)	36 (27.7%)	-
	1	71 (54.6%)	2 (1.5%)	2 (1.5%)	0	28 (21.5%)	42 (32.3%)	42 (32.3%)	64 (49.2%)
	2	41 (31.5%)	20 (15.4%)	12 (9.2%)	10 (7.7%)	5 (3.8%)	17 (13.1%)	26 (20.0%)	39 (30.0%)
	3	10 (7.7%)	35 (26.9%)	55 (42.3%)	48 (36.9%)	9 (6.9%)	11 (8.5%)	19 (14.6%)	12 (9.2%)
	4	8 (6.2%)	61 (46.9%)	54 (41.5%)	59 (45.4%)	1 (0.8%)	1 (0.8%)	0	15 (11.5%)
	5	-	12 (9.2%)	6 (4.6%)	13 (10.0%)	10 (7.7%)	4 (3.1%)	7 (5.4%)	-
**Controls**	0	-	0	1 (1.0%)	0	66 (66.0%)	76 (76.0%)	62 (62.0%)	-
	1	29 (29.0%)	15 (15.0%)	13 (13.0%)	16 (16.0%)	11 (11.0%)	16 (16.0%)	20 (20.0%)	44 (44.0%)
	2	38 (38.0%)	42 (42.0%)	44 (44.0%)	28 (28.0%)	1 (1.0%)	2 (2.0%)	1 (1.0%)	24 (24.0%)
	3	25 (25.0%)	21 (21.0%)	40 (40.0%)	45 (45.0%)	18 (18.0%)	6 (6.0%)	14 (14.0%)	22 (22.0%)
	4	8 (8.0%)	19 (19.0%)	2 (2.0%)	8 (8.0%)	1 (150%)	0	0	10 (10.0%)
	5	-	3 (3.0%)	0	3 (3.0%)	3 (3.0%)	0	3 (3.0%)	-
** *p* ** **value**		**<0.001 ***	**<0.001**	**<0.001**	**<0.001**	**0.015**	**<0.001**	**<0.001**	**0.059**

**Table 4 jcm-11-01848-t004:** Results of Pentacam and in vivo confocal microscopy (IVCM) analysis in subjects with monoclonal gammopathy (MG) and controls in previous studies [41,42,43,54] and in our present study. n.a.: not analysed; MGUS: monoclonal gammopathy of unknown significance.

Author, Year	Number of Eyes (MG/Controls)	Pentacam; Corneal Stromal Light Scattering	Pentacam; Analysed Annular Zones	IVCM; Stromal Hyperreflective Keratocytes	IVCM; Stromal Matrix Hyperreflectivity	IVCM; Stromal Hyperreflective Spikes
Aragona et al., 2016 [54]	62/40	n.a.	n.a.	decreased (in neoplastic patients, compared to controls and MGUS)	increased	n.a.
Enders et al., 2017 [41]	10/26	increased	0–12 mm	n.a.	n.a.	n.a.
Busch et al., 2017 [42]	20/10	increased	0–6 mm	n.a.	n.a.	n.a.
Ichii et al., 2019 [43]	60/64	increased	0–6 mm	n.a.	n.a.	n.a.
Present study	130/100	increased	0–12 mm	increased	n.a.	Increased

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
