# Peer review of "Corneal Densitometry and In Vivo Confocal Microscopy in Patients with Monoclonal Gammopathy—Analysis of 130 Eyes of 65 Subjects"

_jcm, 2022, doi:10.3390/jcm11071848_

Round 1

Reviewer 1 Report

Authors made a good attempt to explain the importance of "Corneal densitometry and in vivo confocal microscopy in patients
 with monoclonal gammopathy – analysis of 130 eyes of 65 subjects"

authors presented the importance of corneal densitometry and invivo microscopy :  since there were no previous studies to support the importance of corneal densitometry and invivo microscopy related to monoclonal gammopathy, this study has shown that Increased corneal light scattering in the central 10 mm annular zone and increased keratocyte hyperreflectivity may give rise to suspicion of monoclocal gammopathy.

authors did put a good effort in detailed explanation regarding the monoclonal gammopathy.

Reviewer 2 Report

General comments:

This paper uses SI and IVCM to quasi-quantify extra scattering from the cornea related to MG. The main claimed benefit is that the cohort size is larger than previous similar studies. In the conclusion they also claim that “spike” feature observation was novel, but I did not feel this was explored sufficiently within the manuscript yet to support that conclusion.

The main issue I have with the manuscript is that the authors seemingly overclaim the novelty of their work and do not fairly reflect prior literature, especially in the abstract.

I feel that improving the statistical significance of knowledge for this research area would merit publication in JCM. Therefore, I recommend major revisions to give the authors the opportunity to make their narrative fairer on proceeding literature, explore and make clear all details where this manuscript is novel and an improvement over prior literature, and add extra sections (I am thinking regarding “spikes”) and figures. Then I would be happy to re-review.

Specific comments:

Title – ok

Abstract -

First sentence – Quick google search brings up several prior studies using IVCM, in regards to the cornea, for monoclonal gammopathy. I don’t think this sentence is fair on prior literature. Similar for Pentacam, overselling the novelty. Make a bit more precise.

Abstract conclusion – how is does this relate to /novel over prior literature

Introduction –

“have been first described in 1900s” 1 – highlights my point about the abstract. 2 – Do you mean the decade or century? Make clear.

Has any previous study used IVCM and pentacam together? Not clear if this is a separate claim to novelty, or a caveat on your other claim.

“larger cohort of subjects” – this seems to be the main claim of novelty and impact for the manuscript. This should be reflected in the abstract. Also, add table of previous studies showing cohort sizes to defend this.

Methods – No comments at this time.

Results –

Table 3 formatting not clear. It would make more sense as a graph.

Table 4 – Results look fine, and useful “raw” data set for the reader to use. However, an additional graphical representation to get the results over to the reader quicker would be useful, in particular the overlap of the percentage (y-axis) distributions (x-axis = score) for each category between the MG and control groups.

Discussion –

First paragraph – verbose repetition of results section for slit lamp. Does not explain what point these values tell the reader.

“Scheimpflug-based” to “In our study,” this part belongs (if not already in) the introduction, not the discussion (and comparison to these studies) of your results. Probably re-write to make direct comparison with your results.

I don’t take your conclusion that scattering increase for MG is particularly novel, perhaps talk (and defend) about how you have made that conclusion more secure than it was before this manuscript.

“These hypothesis need further evaluation.” – I wouldn’t present it like that. Homogeneity of increased scattering in the stroma only, is strongly supportive that the causing process happens in the stromal tissue and is not non-local (transport dependent).

“no previous study described appearance of hyperreflective spikes” – the fact that this element is novel has not come across in the rest of the manuscript. If it is novel, it probably deserves its own section within the manuscript, with descriptive figures showing these spikes and how to identify them from other features.

“central 10 mm annular zone” – what about the other zones?

Round 2

Reviewer 2 Report

I am happy with the changes, and feel the manuscript is now appropriate for JCM. Well done.